# Comparative Analysis of a Rapid Quantitative Immunoassay to the Reference Methodology for the Measurement of Blood Vitamin D Levels

**DOI:** 10.3390/mps8040085

**Published:** 2025-08-01

**Authors:** Gary R. McLean, Samson Soyemi, Oluwafunmito P. Ajayi, Sandra Fernando, Wiktor Sowinski-Mydlarz, Duncan Stewart, Sarah Illingworth, Matthew Atkins, Dee Bhakta

**Affiliations:** 1School of Human Sciences, London Metropolitan University, London N7 8DB, UK; 2National Heart and Lung Institute, Imperial College, London SW7 2AZ, UK; 3School of Computing and Digital Media, London Metropolitan University, London N7 8DB, UK; 4School of Social Sciences and Professions, London Metropolitan University, London N7 8DB, UK; 5Pathology and POCT Department, Oxford Health NHS Foundation Trust, Oxford OX4 4XN, UK

**Keywords:** vitamin D, deficiency, capillary blood, rapid immunoassay, assay validation

## Abstract

Vitamin D is the only vitamin that is conditionally essential, as it is synthesized from precursors after UV light exposure, whilst also being obtained from the diet. It has numerous health benefits, with deficiency becoming a major concern globally, such that dietary supplementation has more recently achieved vital importance to maintain satisfactory levels. In recent years, measurements made from blood have, therefore, become critical to determine the status of vitamin D levels in individuals and the larger population. Tests for vitamin D have routinely relied on laboratory analysis with sophisticated equipment, often being slow and costly, whilst rapid immunoassays have suffered from poor specificity and sensitivity. Here, we have evaluated a new rapid immunoassay test on the market (Rapi-D & IgLoo) to quickly and accurately measure vitamin D levels in small capillary blood specimens and compared this to measurements made using the standard laboratory method of liquid chromatography and mass spectrometry. Our results show that vitamin D can be measured very quickly and over a broad range using the new method, as well as correlate relatively well with standard laboratory testing; however, it cannot be fully relied upon currently to accurately diagnose deficiency or sufficiency in individuals. Our statistical and comparative analyses find that the rapid immunoassay with digital quantification significantly overestimates vitamin D levels, leading to diminished diagnosis of vitamin D deficiency. The speed and simplicity of the rapid method will likely provide advantages in various healthcare settings; however, further calibration of this rapid method and testing parameters for improving quantification of vitamin D from capillary blood specimens is required before integration of it into clinical decision-making pathways.

## 1. Introduction

Vitamin D is the name given to a group of structurally related compounds that are known as secosteroids, a subclass of steroids having a “broken ring” key structural feature. The vitamin was discovered in 1922 when identifying the childhood dietary deficiency that led to rickets [1]. They are fat-soluble molecules, functioning via a nuclear receptor in target cells. It is now recognized that most cells have a vitamin D receptor and that engagement leads to gene expression changes that trigger numerous biological functions in health (cell growth regulation, immune function modulation, and cardiovascular health) alongside the major effects of increasing calcium, magnesium, and phosphate intestinal absorption, whilst supporting bone health [2,3]. Five forms of vitamin D exist (D1–D5), with the most important for human biology being vitamin D2 (ergocalciferol) and vitamin D3 (cholecalciferol). Cells in the liver convert cholecalciferol to calcifediol (25-hydroxycholecalciferol), while ergocalciferol is converted to ercalcidiol (25-hydroxyergocalciferol). These metabolites of vitamin D are collectively referred to as 25-hydroxyvitamin D or 25(OH)D, and can be measured in serum/blood to determine individual and population vitamin D status [4].

Unlike all the other vitamins, vitamin D is only conditionally essential as skin synthesis occurs following UV-B exposure, and dietary vitamin D is considered an additional source. Thus, cholecalciferol is generated in the skin epidermis following a photochemical reaction with provitamin D3 and UV-B radiation from sunlight [5]. Numerous factors, such as climate, urban living, clothing, and skin pigmentation, can reduce sunlight exposure and, therefore, limit cholecalciferol synthesis, placing a greater emphasis on dietary sources to avoid deficiencies. Globally, vitamin D is estimated to be severely deficient (defined as <30 nmol/L) in 5.9% of the US population, 7.4% of Canadians, and 13% of people inhabiting Europe [6]. Those with insufficient vitamin D levels below 50nmol/L are even higher, at 24% in the US, 37% in Canada, and 40% in Europe [6]. These values can vary significantly with age—lower vitamin D levels are often found in young and elderly people, or with regional ethnicity—Caucasian Europeans usually show less deficiency than non-white individuals do. Asian countries such as India, Pakistan, and Afghanistan have reported very high rates > 20% of deficiency (vitamin D < 30 nmol/L), as has Tunisia in Africa. Based on population numbers, in India, this can indicate as many as 490 million individuals with insufficiency [6]. In the UK, due to low sunlight levels in the winter months, vitamin D deficiency is estimated at around 1 in 6 adults; however, Asian and Black populations have even higher rates, although levels can be improved and deficiency symptoms alleviated by daily supplementation [7]. Thus, factors such as ethnicity, where Asian and Black individuals have a higher proportion of vitamin D deficiency, geographic location, and reductions in sunlight levels influencing vitamin D levels, and age, whereby those either young or older have a higher propensity for vitamin D deficiency, are noted risk factors [7]. Male sex has been considered an additional risk factor in the UK [7]. Moreover, in a US study, women were at a higher risk of vitamin D deficiency [8]. It seems likely then that many factors such as age, ethnicity, sex, geography, and season, sun-protective behaviors, lower BMI, lower socioeconomic status (SES), drinking alcohol, and lower milk consumption can influence vitamin D deficiency and insufficiency. There remains, however, some controversy on the cut-off levels for vitamin D deficiency and sufficiency, questioning many of these figures [9]. Nevertheless, epidemiologic evidence and prospective studies have linked vitamin D deficiency with increased risk of many chronic diseases, cardiovascular disease, deadly cancers, type 2 diabetes, and infectious diseases [2]. Thus, a reliable and quick diagnostic test for vitamin D levels in serum/blood would be advantageous to understand population deficiencies and the precise need for further supplementation to maintain 25(OH)D levels at appropriate levels.

The standard reference method for total 25(OH)D measurement in biological samples is high-performance liquid chromatography coupled with tandem mass spectrometry [10]. This method separately quantitates 25(OH)D2 and 25(OH)D3 extremely accurately and with high specificity. The additional use of stringent reference reagents ensures the method meets analytical performance criteria such that measurement results are appropriate to assess the measurement accuracy of clinical vitamin D tests. Due to the relatively low levels of vitamin D, serum is the usual required specimen for measurement. However, the use of dried blood spots (DBS) in combination with liquid chromatography is being used more frequently and is considered a dependable method with reduced sample processing [11]. Immunoassay methods of vitamin D measurement have suffered from poor reliability (for review, see [12]). Specifically, studies have shown that the DiaSource total 25OHD ELISA and IDS RIA provide varied results, while the bioMérieux 25OHD vitamin D Total, DiaSorin Liaison Total, and Euroimmun ELISA provide results within 25% of the expected value [13]. Ideally, a rapid, reliable, and sensitive vitamin D measurement assay that does not rely on laboratory analysis and is also applicable to small volumes of blood specimens, not requiring additional processing, would be extremely useful. Our current investigation has evaluated an immunoassay that is operational with small capillary blood samples, providing qualitative measurements of vitamin D levels in just 15 min without the need for sample processing and laboratory analysis. To extrapolate this lateral flow assay to quantitative measurement, an imaging system is required. Therefore, in this study, we have connected the rapid immunoassay method to IgLoo reader quantification and compared the data obtained to that of the laboratory reference validated method using DBS sampling obtained at the same time as capillary blood sampling. Our results suggest the potential for such an assay to inform the vitamin D status in individuals and populations very quickly.

## 2. Materials and Methods

### 2.1. Study Volunteers

Subjects for the study were recruited from the student and staff body within the School of Human Sciences (SHS) at London Metropolitan University (LMU) on 22nd and 23rd May 2024. Ethical approval was obtained from the SHS ethics panel of LMU. Volunteers completed an online questionnaire with personal details, age, sex, ethnicity, and vitamin D supplementation status. Body mass and height measurements were recorded from each volunteer, and specimens of capillary blood for the two vitamin D assays were collected. One was used immediately in the rapid assay, and the other was retained as DBS for subsequent analysis. A summary of the study’s demographic data is shown in Table 1.

### 2.2. Capillary Blood Vitamin D Test—Rapi-D and IgLoo Reader

The rapid vitamin D test with IgLoo reader quantification was performed immediately with 20 μL of capillary blood according to the manufacturer’s instructions using the Rapi-D quantitative vitamin D test (Affimedix Inc., Hayward, CA, USA). Test batches used were as follows: Ref# 115Q-25, Lot# 23010355, Expiry July 2024. According to the manufacturer, the test has a detection range of 7.5–250 nmol/L, accuracy of 98%, specificity of 100%, and detects both vitamin D2 and D3. In addition, the test has been calibrated against LC-MS/MS and with an R^2^ of 0.96. https://affimedix.com/rapi-drapid-vitamin-d-quantitative-test/ (Accessed on 12th May 2025).

Lateral flow tests were quantified using a calibrated IgLoo Reader 1st generation (Dx365 GmbH, Berlin, Germany), Cat# E0AC2043000096. IgLoo determined vitamin D levels by connecting with Dx Care software, version 1.7.5.879. https://support.dx365.world/care, (Accessed on 12 May 2025).

### 2.3. Dried Blood Spot Vitamin D Testing—Laboratory Reference Method

Additional capillary blood specimens (3–4 spots) were collected and stored on cards as dried blood spots (DBS) for future laboratory analysis by liquid chromatography tandem mass spectrometry (LC/MS-MS). This validated method is closely aligned to serum results also produced using LC-MS/MS [14], and this reference method was used to calibrate the Rapi-D test batches used. The DBSs were sent to Black Country Pathology Services NHS for analysis, with data returned by email. (Address: Clinical Biochemistry Department, Black Country Pathology Services, City Hospital Birmingham, B18 7QH, UK) https://www.vitamindtest.org.uk/, (Accessed on 12th May 2025).

### 2.4. Data Analysis and Statistics

Data were analyzed and presented using GraphPad Prism version 10.3.1 (GraphPad Software LLC, Boston, MA, USA) and IBM SPSS Statistics V. 30 (IBM, Armonk, NY, USA). Statistical analyses (T-test and Spearman’s rho) were performed within each application according to internal software instructions.

## 3. Results

Overall, 50 volunteers participated in the study, and we obtained paired vitamin D measurements from 48 of them, 56% of whom were female, with an average age of 34.2 years, and from multiple diverse ethnicities. The average body mass index of the group was 24.38 kg/m^2^, reflecting that, on average, the group had a healthy weight. Moreover, 48% of volunteers reported supplementing with vitamin D in the month prior to the study. The average blood vitamin D levels of the entire group of volunteers estimated using the rapid test (IgLoo) were 56.2 nmol/L, and using the DBS and LC/MS-MS were 37.4 nmol/L (Table 1). These averages demonstrate that rapid testing with the IgLoo reader recorded, on average, an increase of 18.8 nmol/L above that of DBS. Despite the relatively high standard deviation values, these means are significantly different statistically, with *p* = 0.000238 according to an independent T-test, indicating that the rapid test with IgLoo reader is overestimating vitamin D levels compared to the laboratory reference method of DBS and LC/MS-MS.

To directly compare vitamin D levels and interpretations of status determined independently by IgLoo and DBS, individual raw data were compared, assigned a vitamin D status level according to accepted guidelines, and correlations and bias were determined. Vitamin D status of individuals was interpreted and reflected as deficient, insufficient, and sufficient according to the determined levels by each method. These data are shown in Table 2 and Appendix A. IgLoo provided vitamin D levels with units of ng/mL, and these were converted to nmol/L through multiplication of the value by 2.5. A direct comparison of the differences between vitamin D levels measured by the two methods is also shown in Appendix A. On average, IgLoo generated readings that were 18.8 nmol/L (62%) higher than DBS. Three criteria interpretation of these data according to published UK National Institute for Health and Care Excellence (NICE) guidelines (https://cks.nice.org.uk/topics/vitamin-d-deficiency-in-adults/, (Accessed on 12 May 2025)) was also performed. The 3-criteria interpretation was <25 nmol/L deficient, 25–50 nmol/L insufficient, and >50 nmol/L sufficient. Assigning vitamin D status to these criteria showed there was corroboration between IgLoo and DBS on 28 of the 48 readings, 58.3% similarity (Table 2). However, when looking at deficient samples (bold text), just six (46%) were detected using IgLoo compared to 13 as identified by the laboratory reference method using DBS (Table 2). For further analyses shown in Figure 1, a comparison of the 3-criteria interpretation to a simpler 2-criteria interpretation (<50 nmol/L insufficient, >50 nmol/L sufficient) was performed. Here, vitamin D sufficiency was generally overrepresented whilst deficiency was largely underrepresented with IgLoo measurement (Figure 1A vs. Figure 1B upper panels). The correlation graph (Figure 1A lower panel) demonstrates partial overlap between IgLoo and DBS measurements of vitamin D based on a 3-criteria interpretation and assignment from DBS levels. However, these data do not partition as effectively at the lower end of measurement as compared to that at higher vitamin D levels. Data comparison using the less stringent 2-criteria interpretation (< or >50 nmol/L) revealed there was an improved correlation of 75% (36/48) between IgLoo and DBS, while sufficiency was again overrepresented by IgLoo (Figure 1B upper panel). The correlation graph (Figure 1B lower panel) again showed partial overlap of assigned vitamin D levels, particularly at the higher levels recorded with IgLoo that were not defined as sufficient when measured using DBS. These data are also summarized in Appendix A.

We next performed a correlation analysis on the findings of 48 volunteers for both methods. In Figure 2A, the correlation scatter plot reveals a consistent relationship across the range of vitamin D levels recorded between 10 and 100 nmol/L. A Pearson correlation coefficient (r) of 0.911 between IgLoo and DBS methods of vitamin D measurement was found, identifying that there is a good positive correlation between the two readings obtained by the different methods. This correlation was statistically significant with a *p* value of <0.001 by Spearman’s rho, and the line of best fit has a slope of 1.11. The y-intercept was 14.53, which indicates the IgLoo overestimation.

To further determine the agreement between IgLoo and DBS for vitamin D measurement, these data were analyzed using a Bland–Altman plot (Figure 2B). On average, IgLoo measurements were 18.8 nmol/L (62%) higher than DBS; the 95% limits of agreement (LOA) were from –8.05 to 45.6 nmol/L, with results falling outside, identified as outliers. The greatest similarities between readings are observed at the lower end of measurement (<40 nmol/L), whereas the greatest bias was found at the higher end of the measurement (>70 nmol/L).

These data indicate that rapid measurement of capillary blood vitamin D by lateral flow assay and IgLoo quantification, while somewhat reliable, does not fully represent and reproduce that achieved with DBS through LC-MS/MS. Thus, recalibration of the IgLoo detection software to reflect this lack of concordance is likely necessary.

## 4. Discussion

In this study, we have compared the blood vitamin D levels of 48 volunteers measured by rapid immunoassay and IgLoo quantification to those obtained with the laboratory reference method of measurement using liquid chromatography and tandem mass spectrometry. We established that there was a good correlation (r = 0.91) of vitamin D measurement between the two methods. However, generally inflated values were found with rapid immunoassay and IgLoo digitisation, such that an over-representation of sufficient vitamin D levels and a corresponding under-representation of deficient vitamin D levels was obtained. The Bland–Altman plot analysis demonstrated that IgLoo measurements were higher than DBS. Importantly, reduced bias between IgLoo and DBS was observed at the lower end of measurement (<40 nmol/L), where the cutoff for vitamin D deficiency is found. However, under the current calibration conditions of this assay, IgLoo would not be considered suitable for widespread clinical application, although the significant time improvement and simplicity of specimen collection/processing gained with this method are encouraging. As a tool for large-scale and quick screening of the population for vitamin D levels, IgLoo could provide significant utility, provided regular calibration of the device and the assay itself to quality standards was routinely performed.

Assays for vitamin D have routinely struggled to adapt to small volumes of capillary blood as specimens due to the reduced levels of the vitamin available. Also, adapting reliable assays for vitamin D to analytical methods other than HPLC had previously proved difficult and fraught with problems such as poor antibody specificity, cross reactivity, and confounding matrix substances within the specimens used [15,16]. A relatively recent study compared measurements using a chemiluminescence assay between venous blood specimens and plasma after conversion of the small capillary blood volume to plasma using an anticoagulant and demonstrated very good correlations, linearity, and limits of detection near just 4 nmol/L [17]. These assays still required laboratory analyzers but demonstrated that capillary blood was a reliable specimen for vitamin D measurement. Rapid immunoassays for vitamin D have been developed in recent years, showing good correlation with laboratory reference methods; however, they rely upon serum specimens and fluorescence immunoassay [18]. Therefore, there is a need for assays that combine rapid quantification, small volumes of specimens, simplicity, and analytical reliability.

The widespread use of gold nanoparticle-based lateral flow immunoassays during COVID-19 thrust the technology forward for the diagnosis of SARS-CoV-2 viral infection [19]; however, it still required a subjective element rather than the quantitative nature required for the accurate determination of analyte levels within biological specimens. Thus, lateral flow assays have lagged behind regarding quantification methodology, although the advent of dedicated reader devices [20], including smartphones [21], has improved this aspect recently. The Affimedix vitamin D lateral flow immunoassay (Rapi-D) has become available in recent years, claiming to be the “world’s first rapid sandwich immunochromatographic test for quantitative determination of 25-OH vitamin D in human finger-prick blood”. The manufacturer further states that the assay provides a preliminary diagnostic test result that can be used for the screening of vitamin D deficiency, but recommends further testing of samples using LC-MS/MS to confirm test results. When combined with the RapiRead CUBE Reader device (previously validated with a SARS-CoV-2 neutralizing antibodies lateral flow assay [22]) to convert lateral flow signals to vitamin D levels using a standard curve, the assay becomes quantitative with a sensitivity of 7.5 nmol/L and detection up to 250nmol/L. To our knowledge, no comparative studies evaluating the Rapi-D assay and RapiRead have been independently peer-reviewed and published, although a comprehensive new study appeared recently that evaluated an Affimedix-manufactured ELISA for vitamin D [23]. It is entirely possible that the same or similar antibody reagents are used for this ELISA and the Rapi-D assay, going some way towards validation of the technology. Furthermore, another review of numerous vitamin D assays in comparison to LC-MS/MS stated that “the Affimedix Rapi-D 25OHD Test, Affimedix Micro-D Test and Affimedix Chemi-D Test seemed to deliver measurements most accurately, according to the target value” [13]. Nevertheless, our current study is likely to be the first to directly and comprehensively compare the Rapi-D immunoassay to laboratory reference methodology, albeit using another reader device (IgLoo). Likewise, the IgLoo reader, developed by Experiment X, has not undergone independent verification and the peer-reviewed publication of the data obtained, despite having been applied to quantify numerous lateral flow tests.

We found that the vitamin D assay using capillary blood sampling followed by quantitation by IgLoo reader is a simple and fast assay that produces measurements of vitamin D in 15 min. The software interface of IgLoo to PC-based software provides a robust database of exportable parameters for further analysis. The assay itself provided good measurements that correlated well with the laboratory reference method; however, the bias was generally too high under the current IgLoo calibration conditions, resulting in truly deficient patients being missed, as seen in the findings of this study. For the assay and IgLoo to be adopted into clinical use, more accurate measurements are necessary, particularly at the lower end of detection, where clinical decisions regarding vitamin D deficiency are made. Our data did show that the greatest similarity in measurements was seen at this lower end of detection, but that vitamin D deficiency was nevertheless underestimated by IgLoo with a corresponding overestimate of the number of subjects with sufficient levels. Improvements to the calibration of IgLoo for vitamin D measurement are therefore necessary before its reliability can be fully established.

Whilst using IgLoo for the rapid measurement of vitamin D levels provides many advantages over laboratory-based measurements, the adoption of this technology, as it currently stands in interpreting vitamin D deficiency and, therefore, related downstream healthcare applications, requires additional investigation and improved confidence in assigning correct values. Furthermore, this exploratory study with n = 48 is relatively small and difficult to draw full conclusions from. Therefore, further testing with an increased number of samples following recalibration of the IgLoo and rapid vitamin D test is recommended.

## Figures and Tables

**Figure 1 mps-08-00085-f001:**
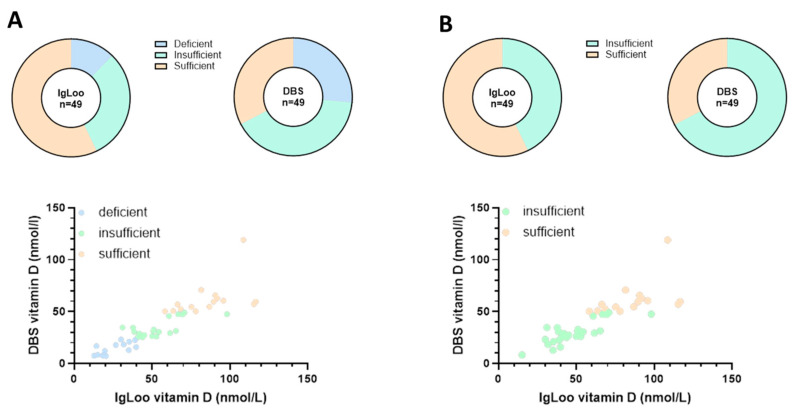
Graphical representation and interpretation of vitamin D levels. Levels of vitamin D obtained by both IgLoo and DBS were analyzed using 3-criteria (**A**) or 2-criteria (**B**) interpretation.

**Figure 2 mps-08-00085-f002:**
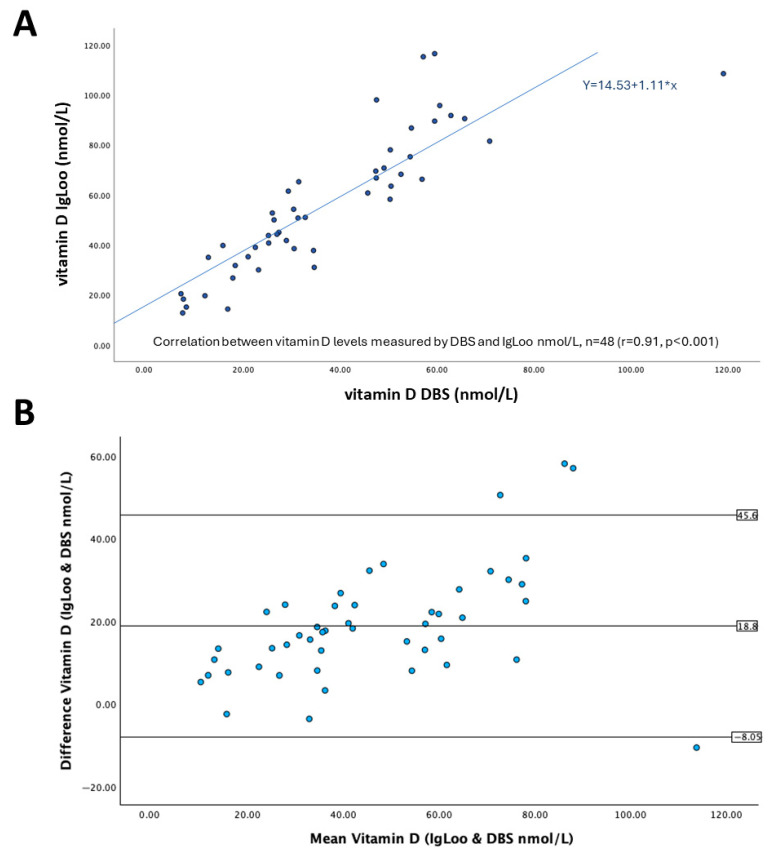
Correlation analysis (**A**) and Bland–Altman plot (**B**) for vitamin D levels obtained using DBS and IgLoo.

**Table 1 mps-08-00085-t001:** Demographics of volunteers for the determination of blood vitamin D levels.

Total Number of Participants, *n*	48
Males (%)	21 (44)
Females (%)	27 (56)
Mean age in years (SD)	34.2 (8.99)
Vitamin D supplementation: number of volunteers that report taking dietary supplements within the last month prior to sampling (%)	24 (48)
Ethnicity, self-reported, n (%)	Asian 19 (40)
	Black 12 (25)
	White 13 (27)
	Hispanic 1 (2)
	Arab 1 (2)
	Mixed 2 (4)
Mean body mass index kg/m2 (SD)	24.38 (4.57)
Mean vitamin D status nmol/L *	
Rapid test with IgLoo reader (SD)	56.2 (27.4)
DBS and LC/MS-MS (SD)	37.4 (21.4)

* Volunteers provided capillary blood samples (20 μL) for immediate rapid immunoassay vitamin D testing and similar volumes of capillary blood to prepare 3 to 4 dried blood spots (DBSs) for subsequent laboratory analysis of vitamin D using LC/MS-MS.

**Table 2 mps-08-00085-t002:** Raw data and interpretation of vitamin D status.

Subject ID	IgLoo Vitamin D Levels from Reader (ng/mL)	IgLoo Vitamin D Levels (nmol/L)	IgLoo Vitamin D Levels Interpretation ^a^	DBS Vitamin D Levels (nmol/L)	DBS Vitamin D levels Interpretation ^a^
0001	12.4	31	Insufficient	34.65	Insufficient
0002	36.2	90.5	Sufficient	65.7	Sufficient
0003	7.86	19.65	**Deficient**	12.1	**Deficient**
0004	17.5	43.75	Insufficient	25.2	Insufficient
0005	20.3	50.75	Sufficient	31.3	Insufficient
0006	7.32	18.3	**Deficient**	7.65	**Deficient**
0007	24.3	60.75	Sufficient	45.7	Insufficient
0008	39.2	98	Sufficient	47.5	Insufficient
0009	35.8	89.5	Sufficient	59.5	Sufficient
0010	16.3	40.75	Insufficient	25.25	Insufficient
0011	18	45	Insufficient	27.35	Insufficient
0012	26.7	66.75	Sufficient	47.45	Insufficient
0013	8.18	20.45	**Deficient**	7.15	**Deficient**
0014	31.2	78	Sufficient	50.35	Sufficient
0015	25.4	63.5	Sufficient	50.5	Sufficient
0016	36.7	91.75	Sufficient	62.85	Sufficient
0017	5.73	14.325	**Deficient**	16.8	**Deficient**
0018	10.7	26.75	Insufficient	17.85	**Deficient**
0019	21.7	54.25	Sufficient	30.4	Insufficient
0020	15.1	37.75	Insufficient	34.5	Insufficient
0021	23.3	58.25	Sufficient	50.3	Sufficient
0022	32.6	81.5	Sufficient	70.85	Sufficient
0023	21.1	52.75	Sufficient	26	Insufficient
0024	5.11	12.775	**Deficient**	7.5	**Deficient**
0025	14.1	35.25	Insufficient	21	**Deficient**
0026	15.4	38.5	Insufficient	30.5	Insufficient
0027	12	30	Insufficient	23.15	**Deficient**
0028	38.3	95.75	Sufficient	60.55	Sufficient
0029	46.6	116.5	Sufficient	59.5	Sufficient
0030	17.7	44.25	Insufficient	26.95	Insufficient
0031	12.7	31.75	Insufficient	18.35	**Deficient**
0032	27.8	69.5	Sufficient	47.35	Insufficient
0033	24.6	61.5	Sufficient	29.3	Insufficient
0034	15.6	39	Insufficient	22.5	**Deficient**
0035	>150	n.d.	High ^b^	80	Sufficient
0036	28.3	70.75	Sufficient	49.05	Insufficient
0037	26.1	65.25	Sufficient	31.45	Insufficient
0038	34.7	86.75	Sufficient	54.7	Sufficient
0039	43.4	108.5	Sufficient	119.1	Sufficient
0040	30.1	75.25	Sufficient	54.45	Sufficient
0041	20.4	51	Sufficient	32.8	Insufficient
0042	46.1	115.25	Sufficient	57.15	Sufficient
0043	24.8	62	Sufficient	n.m.	n.m.
0044	15.9	39.75	Insufficient	15.8	**Deficient**
0045	14	35	Insufficient	12.8	**Deficient**
0046	20	50	Sufficient	26.35	Insufficient
0047	27.3	68.25	Sufficient	52.55	Sufficient
0048	26.5	66.25	Sufficient	56.9	Sufficient
0049	16.7	41.75	Insufficient	28.9	Insufficient
0050	6.05	15.125	**Deficient**	8.25	**Deficient**

^a^ <25 nmol/L deficient, 25–50 nmol/L insufficient, >50 nmol/L sufficient, according to UK National Institute for Health and Care Excellence (NICE) guidelines. Samples measured as deficient are highlighted in bold. ^b^ Outside the upper detection limit of IgLoo, potentially abnormally high levels. n.d.—not determined, due to IgLoo reading outside detection limit; n.m.—no measurement taken.

## Data Availability

Reasonable requests for data are available by contacting the corresponding authors.

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
