# Peer review of "Comparative Analysis of a Rapid Quantitative Immunoassay to the Reference Methodology for the Measurement of Blood Vitamin D Levels"

_mps, 2025, doi:10.3390/mps8040085_

Round 1

Reviewer 1 Report

Comments and Suggestions for Authors

This manuscript evaluates a new rapid immunoassay method available on the market for measuring vitamin D levels in capillary blood samples, comparing its performance with results obtained from the standard laboratory method using liquid chromatography and mass spectrometry. The study aims to provide a reference for the clinical selection of accurate and rapid methods for vitamin D determination. The following suggestions are offered to help improve the manuscript:

  1. How do the specificity and sensitivity of this method compare with those of the standard reference method (reference [9])? Why was the standard reference method not chosen?

  2. Please include an analysis of whether vitamin D deficiency is associated with factors such as gender, age, and ethnicity.

  3. Ensure consistency in the use of units throughout the manuscript and in the capitalization of the letter "V" in "vitamin D."

Author Response

We appreciate the thoughtful comments, questions, and efforts of all three reviewers, which have contributed significantly to improving our manuscript.

Reviewer 1 inquired about the sensitivity and specificity of the two tests and their comparison. While our data did not allow for direct calculation of these metrics, we have now incorporated manufacturer-provided information into the Methods section (page 3). In the context of Vitamin D measurement, sensitivity refers to the test’s ability to detect low analyte concentrations, typically described by the limit of detection (LOD), limit of quantification (LOQ), and dynamic range. The Affimedix Rapi-D™ Rapid Vitamin D Quantitative Test has a detection range of 7.5 nmol/L to 250 nmol/L, with an LOD of 7.5 nmol/L, and is capable of detecting both Vitamin D2 and D3.

Specificity concerns the test’s ability to distinguish between 25 OH D2 and D3 and to avoid cross-reactivity with other blood components. The LC-MS/MS method excels in this regard due to its analytical precision, whereas immunoassays, which rely on antibody specificity, may be less robust. The Rapi-D test measures both Vitamin D2 and D3.

The standard LC-MS/MS method was chosen as the comparator assay. If the reviewer’s concern pertained to the use of capillary blood versus serum, we note that the reference method has demonstrated comparable performance with both specimen types.

The reviewer also requested analysis of Vitamin D deficiency in relation to gender, age, and ethnicity. Age and ethnicity are discussed in the introduction (page 2), but gender was not included as it is not typically a factor in deficiency rates.

We have standardized all units to nmol/L throughout the manuscript. Where original values obtained from IgLoo were in ng/ml, these have been converted as shown in Table 2 (page 5). We have also ensured consistent capitalization of “Vitamin D.”

Reviewer 2 Report

Comments and Suggestions for Authors

This paper describes a comparison of a rapid immunoassay for vitamin D with the reference method (LC MS/MS).

I have some major comments.

The authors mention poor reliability of the immunological tests in the introduction (ref. 11). No discussion is made about the difference between the antibodies they use and the antibodies used in previous immunological methods. Where is the originality? The speed of concentration determination? Is this sufficient for an accurate diagnosis?

I recreated Figure 2A with the data provided by the authors. I find y = 0.6812x - 0.9139 R² = 0.7595, not what the authors show (y = 14.53 + 1.11x). The slope cannot be 1; the curve clearly highlights an overestimation of Igloo compared to LC MS. What values ​​did the authors use to obtain this graph?

I don't really understand what this test is: "rapid immunoassay method and Igloo reader quantification"? Is it an immunological test coupled with a reader? Why is this test rapid?

The immunological test is not validated, or at least the authors don't show any validation.

Can we compare results between a non-validated test and a reference method?

More precisely:

- P3 in the Capillary Blood Vitamin D Test section: 20 µL and not 20 l

- Concentration units should be included in Table 1

Author Response

We appreciate the thoughtful comments, questions, and efforts of all three reviewers, which have contributed significantly to improving our manuscript.

Reviewer 2 raised concerns about the lack of discussion on antibodies used in immunoassays and their potential impact on test discrepancies. While this is a valid point, detailed antibody information is proprietary and not disclosed by manufacturers. We acknowledge that antibody quality likely affects assay performance and have addressed this in the discussion. Without specific antibody data, further analysis is not feasible, though we anticipate that newer assays with improved antibodies will yield better results.

The reviewer also questioned the originality of our work and whether speed alone suffices for accurate diagnosis. Our manuscript emphasizes the need for rapid yet accurate diagnostics, and our study directly compares the rapid assay to the gold standard. Our findings indicate that the rapid assay requires further calibration, as noted in the discussion.

Regarding concerns about Figure 2A and statistical analyses, we have reanalyzed the data independently and obtained consistent results. We believe discrepancies may stem from unit conversion errors. All analyses used nmol/L, and we are confident in the accuracy of our plot and analysis. The IgLoo measurement’s tendency to overestimate Vitamin D levels is reflected in the y-intercept, as stated.

We have clarified statements about the rapid Vitamin D assay, especially in the introduction (page 3), to address any confusion. We thank the reviewer for highlighting this need for clarification.

Finally, the reviewer questioned the validity of comparing non-validated to validated tests. Our study serves as a pilot validation, aligning the rapid test with an NHS-validated assay. This independent validation supports the potential adoption of a rapid POCT for Vitamin D, which could streamline population-level deficiency screening.

Minor formatting issues, including the loss of “ul” and unit consistency in Table 1, have been corrected.

Reviewer 3 Report

Comments and Suggestions for Authors

In my opinion, there is no scientific novelty in the work. The Authors stated that the Rapi-D & IgLoo Reader test is already on the market, which means that its effectiveness in determining vitamin D has been assessed earlier, because otherwise it would not be available on the market. So why have the Authors additionally checked its usefulness in determining vitamin D - there is no appropriate explanation in the Introduction?

„ Vitamin D from capillary blood specimens is required before integration of it into clinical decision-making pathways measured very quickly and over a broad range by the new method, correlate relatively well with standard laboratory testing, however cannot be fully relied upon currently to accurately diagnose deficiency or sufficiency in individuals” I would like the Authors to provide examples of what tests the Authors mention (specific names of these tests) that give incorrect results/correlations, and please also provide references to this information.

The study population of 48 is not sufficient to draw conclusions

Has the HPLC-MS method been validated? Complete the validation parameters.

Has the Rapi-D & IgLoo assay been validated?

The literature is very outdated and there are no current references, publications from 1987 or 1997 are unnecessarily cited.

Author Response

We appreciate the thoughtful comments, questions, and efforts of all three reviewers, which have contributed significantly to improving our manuscript.

Reviewer 3 questioned the scientific novelty of the work. We have expanded the introduction to underscore the importance of rapid Vitamin D measurement and the necessity of comparison studies with gold standard methods for healthcare adoption.

The reviewer requested more details on tests that have produced unreliable results, along with supporting references. We have addressed this in the third paragraph of the introduction and added a relevant reference.

Regarding the sample size (“n=48”), we acknowledge that this is a small-to-moderate number for statistical evaluation. While sufficient for a pilot or exploratory study, larger sample sizes are needed for more robust conclusions. We have added text near the end of the discussion to address this limitation and note that a larger study is planned.

On the question of validation for the Rapi-D and IgLoo assays, we clarify that our study aims to provide this validation, as now emphasized in the revised text.

Finally, the reviewer raised concerns about outdated references. Our citations aim to reflect both the historical context and current developments in Vitamin D testing. Most references are from the past decade, with three from 2025, ensuring the manuscript is up to date. The inclusion of a few older, foundational references does not detract from the relevance or currency of our work.

Round 2

Reviewer 1 Report

Comments and Suggestions for Authors

The authors claim that age and ethnicity are discussed on the second page of the introduction; however, these factors are not clearly elaborated. Furthermore, the authors state that gender is not a typical factor influencing vitamin D deficiency, whereas existing literature, such as "Prevalence, trend, and predictor analyses of vitamin D deficiency in the US population, 2001–2018", has demonstrated that gender does have a significant impact.

Author Response

We thank the reviewer for alerting us to the lack of clarity of the section of the introduction explaining risk factors and demographics of Vitamin D deficiency and insufficiency. We have now added 9 lines of text in the appropriate place of the introduction explaining this further and also the supporting reference shared by the reviewer that indicates sex is a factor associated with Vitamin D deficiency. This should now clarify the many risk factors associated with Vitamin D deficiency and insufficiency and satisfy the reviewers concern.

Reviewer 2 Report

Comments and Suggestions for Authors

I recreated Figure 2A with the data provided by the authors. I find y = 0.6812x - 0.9139 (x = Igloo and y = LC MS/MS ) R² = 0.7595, not what the authors show (y = 14.53 + 1.11x). The slope cannot be 1; the curve clearly highlights an overestimation of Igloo compared to LC MS. What values ​​did the authors use to obtain this graph?

Author Response

The reviewer has very carefully analysed the data shown in Fig 2A and quite rightly points out where discrepancies may exist. However, the reviewer has restated their earlier criticisms without addressing our rebuttal to that and in our opinion has not performed the analysis entirely correctly as we describe below. 

  1. The reviewer has not addressed our rebuttal to their criticism of the data used and plot shown as Fig 2A but has merely restated their original contention that the plot is incorrect based on their analysis without showing their detailed evidence and data used.
  2. We have used the data shown in Table 2 columns 3 (IgLoo Vitamin D levels nmol/L) and 5 (DBS Vitamin D levels nmol/L) and contend that the reviewer may have not used these or mixed units.
  3. Independent verification of the plots and statistical analyses was recently performed by different authors several times and also by using two different software packages, SPSS and GraphPad Prism. On each occasion, we achieved the same or very similar plot pattern, line of best fit, line equation and correlation coefficient (r=0.91).
  4. The y-intercept cannot be negative as the reviewer suggests since IgLoo was found to overestimate Vitamin D levels, as stated throughout the paper and therefore should be a positive value. In fact, this difference can be explained by the fact that the reviewer has inverted the axes, we plot IgLoo data on the y-axis and DBS on the x-axis - they have done the opposite. So the reviewer equation and analysis is based on inverting the plot compared to ours.
  5. The slope is not 1 as is stated by the reviewer but is calculated as 1.11 according to the line equation and the reviewer slope is different due to inverting the axes as explained in point 4.
  6. Lastly, our software analysis has determined r and not R2 which are different coefficients and can therefore explain why the numbers obtained are different.

Therefore we stand by the original Figure and the analysis done. Ultimately, this does not alter the findings and message of the paper. i.e. IgLoo is good and has advantages other other methodology but does not currently reproduce the measurements made by LC-MS/MS well enough to be adopted into clinical pathways.

Reviewer 3 Report

Comments and Suggestions for Authors

I do not have any additional comments. It can be published in presented form

Author Response

We thank the reviewer for approving this version of the paper.